# Cross-Linked Luminescent Polymers Based on *β*-Diketone-Modified Polysiloxanes and Organoeuropiumsiloxanes

**DOI:** 10.3390/polym14132554

**Published:** 2022-06-23

**Authors:** Eleonora E. Kim, Yuriy N. Kononevich, Yulia S. Dyuzhikova, Dmitry S. Ionov, Dmitry A. Khanin, Galina G. Nikiforova, Olga I. Shchegolikhina, Viktor G. Vasil’ev, Aziz M. Muzafarov

**Affiliations:** 1A.N. Nesmeyanov Institute of Organoelement Compounds, Russian Academy of Sciences, 119991 Moscow, Russia; ela-kim@mail.ru (E.E.K.); vysochinskaja@yandex.ru (Y.S.D.); d.a.khanin@gmail.com (D.A.K.); ggn@ineos.ac.ru (G.G.N.); olga@ineos.ac.ru (O.I.S.); viktor@ineos.ac.ru (V.G.V.); 2Photochemistry Center, FSRC “Crystallography and Photonics”, Russian Academy of Sciences, 119421 Moscow, Russia; dmitriy.ionov@gmail.com; 3N.S. Enikolopov Institute of Synthetic Polymeric Materials, Russian Academy of Sciences, 117393 Moscow, Russia

**Keywords:** cross-linked polysiloxanes, metal-ligand interaction, *β*-diketone, luminescence

## Abstract

Nowadays, luminescent materials attract wide attention due to their valuable characteristics and broad area of potential application. Luminescent silicone-based polymers possess unique properties, such as flexibility, hydrophobicity, thermal and chemical stabilities, etc., which allow them to be utilized in various fields, such as optoelectronics, solid-state lasers, luminescent solar concentrators, sensors, and others. In the present work, a metal-ligand interaction approach was applied to obtain new cross-linked luminescent polymers based on multiligand polysiloxanes with grafted *β*-diketone fragments and organoeuropiumsiloxanes containing various organic substituents. Organoeuropiumsiloxanes were utilized as a source of Eu^3+^ ions due to their compatibility with the silicon matrix. All synthesized polymers were fully characterized and their physicochemical, mechanical, self-healing, optical, and thermal properties were studied.

## 1. Introduction

Polymers cross-linked by metal-ligand interactions are modern materials that find applications in various fields, as they can possess such valuable properties as self-healing [1,2,3,4,5,6], stimulus-responsiveness [7,8,9], shape-memory [10], and conducting properties [11]. Networks double cross-linked by coordinative and hydrogen bonds can be used as functional binders for high-performance silicon submicroparticle anodes [12]. There are also specific fields devoted to gels constructed through metal-ligand coordination with improved mechanical properties [13,14,15]. Introducing lanthanide ions into the discussed polymer matrices leads to luminescent materials with valuable optical properties.

There are some routes for obtaining luminescent materials, namely, the incorporation of luminophores into a polymeric matrix, either organic fluorescent dyes [16,17,18,19] or inorganic dyes, such as quantum dots [20,21], nanoparticles [22,23,24], lanthanide salts [25], MOFs [26], and complex compounds [27,28,29,30]. Luminophores can be covalently bonded to a matrix [31,32] or used as fillers [20,21,23,24,26,28]. Such polymer materials are mainly supposed to be utilized as light-emitting diodes (OLEDs) [18,20,23], including white OLEDs [21,29], but also as luminescent coatings [26], chemosensors [19], and upconverting composite materials [24]. Lanthanide ionic liquid/polymer composites show an excellent adsorption capacity for particulate matter in air pollution [33]. Polymers modified with organic fluorescent dyes can be utilized as composite materials with mechanochromic and thermally responsive properties [34]. Cross-linking by the metal-ligand interaction route seems to be very efficient for the development of polymeric materials with luminescent properties, since, in this case, metal ions serve both as cross-linking agents and as sources of luminescence that are chemically bonded to polymer matrix.

Polysiloxanes are widely used polymers owing to their many significant properties, such as high elasticity and flexibility, thermal stability, biocompatibility and low toxicity, superior film-forming ability, and resistance to thermal, chemical, and radiation degradation [35]. These features make polysiloxanes some of the most valuable polymers to be utilized in various scientific and industrial areas. There are many successful examples of preparations of luminescent materials based on various architectures of organosilicon compounds. Thus, linear polysiloxanes, hyperbranched organosilicones, and polyhedral oligomeric silsesquioxane (POSS) derivatives could be utilized for the preparation of unconventional fluorescent materials [36]. Luminescent polysiloxanes can possess self-healing properties [37,38,39] can be applied in photonics [40,41] and as sensing devices for detecting nitroaromatic compounds [42] or oxygen [43,44]. Luminescent complexes could be introduced into a polysiloxane matrix as catalysts for a hydrosilylation-based cross-linking reaction, thereby playing two roles [45,46]. Integrating both Eu^3+^ and Tb^3+^ ions into ligand-terminated oligodimethylsiloxane leads to obtaining luminescent polymer materials with tuned green and red emissions [47]. There are many examples of luminescent silicone materials with different properties based on complexes of lanthanide with numerous ligands [48,49,50,51].

Among the various ligands, *β*-diketones represent one of the most widely used classes for chelating lanthanide ions, especially Eu^3+^. The f–f transitions that result in light emissions from the lanthanides are both spin- and parity-forbidden, which causes their low extinction coefficient and luminescence intensity. Therefore, an indirect excitation is necessary, which is also known as the antenna effect, where a ligand is used as a light-harvesting antenna and subsequent excited-state energy is transferred from the ligand to the central lanthanide ion [52,53]. *β*-Diketones are some of the important “antennas” in terms of their high harvest emissions because of the effectiveness of the energy transfer from this type of ligand to the Eu^3+^ ion. Moreover, *β*-diketones chelate uncharged bidentate ligands, which possess such advantages as a monovalent negatively charged binding site, which results in the formation of neutral 3:1 ligand–lanthanide luminescent complexes [54]. The photophysical properties of lanthanide–*β*-diketonate complexes have been widely investigated [55,56,57].

Inorganic salts are often used as the sources of lanthanide ions [27,28,29,30], but there are some difficulties related to the insolubility of inorganic salts in organic solvents, the necessity of washing off the inorganic counterions, and the low compatibility of inorganic salts with the polymer matrix. Taking these obstacles into account, organometallasiloxanes seem to be more suitable sources of Eu^3+^ ions. Organometallasiloxanes are a unique class of organosilicon compounds containing RSi(O)OM units, where M is a metal ion. A wide variety of metallasiloxanes can be derived from different Si-OH-containing organosilicon compounds [58,59]. Molecules of these compounds differ in the nature of the metal ion used and in their structures—there are cyclic, sandwich, and cage-like compounds, resins, etc. [60,61,62,63,64].

In this work, a route for the preparation of new elastic luminescent polymer materials based on polysiloxanes was developed. Organosilicon polymers cross-linked by an interaction between a *β*-diketone ligand covalently bonded to the polymer chain and Eu^3+^ ions were synthesized and characterized. To avoid difficulties related to the compatibility of inorganic Eu^3+^ salts with polysiloxanes, oligoorganoeuropiumsiloxane resins with two different organic substituents were utilized. The physicochemical, mechanical, optical, and thermal properties of the prepared polymers were studied. In addition, these materials possess self-healing properties and demonstrate sensitivity to ammonia vapors.

## 2. Experimental Methods

### 2.1. Materials

Octamethylcyclotetrasiloxane, hexamethyldisiloxane, Amberlyst 15, PhSi(OEt)_3_, EtSi(OEt)_3_, EuCl_3_ × 6H_2_O, and Eu(NO_3_)_3_ × 6H_2_O were purchased from ABCR (Karlsruhe, Germany). Platinum(0)-1,3-divinyl-1,1,3,3-tetramethyldisiloxane complex solution (in xylene, Pt-2%) was purchased from Sigma-Aldrich (St. Louis, Missouri, US). Methylhydrosiloxane cyclic oligomers (MHSCO) **2** were prepared with a method similar to one described earlier [65]. Allyl-dibenzoylmethane (Allyl-DBM) **5** was synthesized with the method that we previously described [66]. All chemicals were used without further purification. All solvents were purified before use. Toluene and tetrahydrofuran (THF) were distilled from CaH_2_. Acetonitrile was distilled.

### 2.2. Synthesis

#### 2.2.1. Polysiloxane with Distributed Silylhydride Groups (**4**)

Octamethylcyclotetrasiloxane **1** (15 g, 50.6 mmol), MHSCO **2** (0.68 g, 2.5 mmol), hexamethyldisiloxane **3** (0.06 g, 0.3 mmol), and Amberlyst 15 (0.8 g) were stirred at 60 °C for 8 h. Then, the reaction mixture was cooled down to room temperature and dissolved in toluene (150 mL), and Amberlyst 15 was filtered off. Polymer **4** was purified through reprecipitation from toluene by adding acetonitrile. Polymer **4** was obtained as a colorless oil with 65% yield and had the following characteristics: M_n_ = 29,500 Da, M_w_ = 49,600 Da, PDI = 1.68. IR (KBr, cm^−1^): 2969, 2911, 2156, 1448, 1415, 1264, 1096, 1023, 914, 868, 801, 703, 668, 504. ^1^H NMR (400 MHz, CDCl_3_): δ 0.06 (s, 251H, Si-CH_3_), 4.66–4.68 (m, 1H, Si-H).

#### 2.2.2. Polysiloxane with Distributed Dibenzoylmethane Groups (**6**)

A mixture of polysiloxane **4** (5 g, 1.5 mmol), Allyl-DBM **5** (0.63 g, 2.39 mmol), and a 50 μL solution of Karstedt’s catalyst were stirred in dry toluene (50 mL) under an argon atmosphere at 40 °C for 48 h. After the reaction was complete, toluene (50 mL) was added to the reaction mixture, and the product was precipitated by acetonitrile. Polymer **6** was obtained as a brown oil with 95% yield and had the following characteristics: M_n_ = 47,000 Da, M_w_ = 123,000 Da, PDI = 2.61. IR (KBr, cm^−1^): 2963, 2905, 1609, 1412, 1260, 1090, 1019, 865, 801, 688, 500. ^1^H NMR (400 MHz, CDCl_3_): δ 0.06 (s, 270H, CH_3_), 0.57 (t, 2H, *J* = 8.1 Hz, CH_2_), 1.70 (t, 2H, *J* = 8.1 Hz, CH_2_), 2.69 (t, 2H, *J* = 8.1 Hz, CH_2_), 6.84 (s, 1H, CH, COCHCO), 7.28 (d, 2H, *J* = 8.0 Hz, Ar), 7.46–7.56 (m, 3H, Ar), 7.91 (d, 2H, *J* = 8.1 Hz, Ar), 7.98 (d, 2H, *J* = 7.0 Hz, Ar), 16.95 (s, 1H, OH). ^29^Si NMR (79 MHz, CDCl_3_): δ-21.92.

#### 2.2.3. Preparation of Oligophenyleuropiumsiloxane (**7 a**)

A three-necked flask equipped with a magnetic stirrer and a reflux condenser was loaded with n-butanol (39 mL), PhSi(OEt)_3_ (5.33 g, 22.2 mmol), and NaOH (0.89 g, 22.2 mmol). The reaction mixture was stirred under reflux until the solution became transparent, and then boiling was continued for another 20 min. EuCl_3_ × 6H_2_O (2.71 g, 7.40 mmol) was dissolved in n-butanol (74 mL) and then added via a dropping funnel to the solution of sodium *cis*-tetraphenylcyclotetrasiloxanolate, which was obtained in the previous step. Then, the reaction mixture was stirred under reflux for 20 min. The hot solution was then filtered off of the NaCl through a paper filter and left to cool down. After that, the reaction mixture was concentrated in vacuo until a resin formed. The resulting substance was obtained as a white crystalline powder at 73% yield (3.63 g). The elemental analysis calculated (%) for C_26_H_35_Si_3_Eu_1_O_8_: C, 43.87; H, 4.96; Si, 11.84; Eu, 21.35; found: C, 42.72; H, 4.12; Si, 12.55; Eu, 21.15.

#### 2.2.4. Preparation of Oligoethyleuropiumsiloxane (**7 b**)

A three-necked flask equipped with a magnetic stirrer and a reflux condenser was loaded with absolute ethanol (40 mL), EtSi(OMe)_3_ (4.7 g, 31.28 mmol) and NaOH (1.25 g, 31.28 mmol). The reaction mixture was stirred under reflux until the solutions became transparent, and then boiling was continued for another 20 min. Eu(NO_3_)_3_ × 6H_2_O (4.65 g, 10.42 mmol) was dissolved in absolute ethanol (80 mL) and then added via a dropping funnel to the solution of sodium oligoethylsiloxanolate, which was obtained in the previous step. Then, the reaction mixture was stirred under reflux for 20 min. The hot solution was then filtered off of the NaCl through a paper filter and left to cool down. After that, the reaction mixture was concentrated until a resin formed. The resulting substance was obtained as a white crystalline powder at 81% yield (3.98 g). The elemental analysis Calculated (%) for C_28_H_82_Si_10_Eu_3_O_28_: C, 20.97; H, 5.15; Si, 17.51; Eu, 27.93; found: C, 19.58; H, 4.76; Si, 17.9; Eu, 22.4.

#### 2.2.5. Preparation of Europium-Contained Cross-Linked Polymers (**8 a**–**d**)

Solutions of polymer **6** (0.5 g, 0.14 mmol) in THF (10 mL) and **7 a** (0.036 g, 0.035 mmol (Eu^3+^) for **8 a**; 0.072 g, 0.07 mmol (Eu^3+^) for **8 b**) or **7 b** (0.018 g, 0.035 mmol (Eu^3+^) for **8 c**; 0.036 g, 0.07 mmol (Eu^3+^) for **8 d**) in THF (3 mL) were combined and intensively shaken. Then, the obtained mixture was poured out onto a Teflon plate. After the solvent’s evaporation was complete and a film was formed, the latter was placed in a vacuum oven and kept at 80 °C and 1 mbar for 8 h. Cross-linked polymers were obtained as yellow (**8 a**, **c**), grayish (**8 b**), or colorless (**8 d**) films. **8 a**. IR (KBr, cm^−1^): 2962, 2905, 1596, 1414, 1256, 1076, 1006, 863, 785, 696, 661, 497. **8 b**. IR (KBr, cm^−1^): 2962, 2905, 1595, 1413, 1257, 1076, 1006, 863, 784, 697, 660, 501. **8 c**. IR (KBr, cm^−1^): 2963, 2906, 1607, 1416, 1261, 1097, 1020, 866, 799, 701, 660. **8 d**. IR (KBr, cm^−1^): 2969, 2910, 1599, 1417, 1262, 1086, 1018, 868, 795, 702, 665.

### 2.3. Characterization

The ^1^H and ^29^Si NMR spectra were recorded on a Bruker Avance II spectrometer (400 MHz; Billerica, Massachusetts, US. Chemical shifts were reported relative to chloroform (δ = 7.25 ppm) for ^1^H NMR.

IR spectra were recorded on an IR spectrometer with a Fourier transform Shimadzu IRTracer-100 (Kyoto, Japan). KBr pellets and thin layers on KBr windows were used as samples (compounds **4**, **6**, **8 c**,**d**). The FTIR spectra for compounds **7 a**,**b** and **8 a**,**b** were acquired using the ATR mode (QATR™ 10 single-reflection ATR accessory with a diamond crystal).

SEM images were obtained using a Hitachi TM4000PLUS field-emission scanning electron microscope (Tokyo, Japan).

Luminescence spectra were acquired on a Shimadzu RF-6000 spectrofluorophotometer (Kyoto, Japan).

Luminescence decay curves were acquired on a Fluotime 300 spectrofluorometer (Picoquant, Berlin, Germany). An LDH-D-C-375 laser was used as the excitation source (λ_ex_ = 375 nm). Data fitting was performed with the Easytau2 (Picoquant) software by using multiexponential fitting of experimental data.
$$\mathrm{Dec}\left(t\right)=\left[\int_{-\infty }^{t}dt’\left[\mathrm{IRF}\left(t-Shift_{IRF}\right)-Bkgr_{IRF}\right]\left[\overset{n_{Exp}}{\underset{i=1}{\sum }}A_{i}e^{-\frac{t-t^{\prime }}{\tau_{i}}}\;+A_{Scatt}\delta \left(t-t^{\prime }\right)\right]\right]+Bkgr_{Dec}\;I_{m}=A_{m}\tau_{m}\;\tau_{Av\;Int}=\overset{n_{exp}}{\underset{\begin{array}{c} i=1\\ I_{i}>0 \end{array}}{\sum }}I_{i}\tau_{i}/\overset{n_{exp}}{\underset{\begin{array}{c} i=1\\ I_{i}>0 \end{array}}{\sum }}I_{i}\;\tau_{Av\;Amp}=\overset{n_{exp}}{\underset{\begin{array}{c} i=1\\ A_{i}>0 \end{array}}{\sum }}A_{i}\tau_{i}/\overset{n_{exp}}{\underset{\begin{array}{c} i=1\\ A_{i}>0 \end{array}}{\sum }}A_{i}$$

GPC analyses were performed in toluene (1 mL/min) using a Shimadzu Prominent system equipped with an RID-20A refractive index detector (Kyoto, Japan). The GPC columns (Phenogel) were calibrated with polystyrene standards (PSS).

Thermogravimetric analysis (TGA) was performed using a Shimadzu DTG-60H derivatograph (Kyoto, Japan) on samples with a weight of about 5 mg at a heating rate of 10 °C/min in air and argon. The temperature at which a weight loss of 5% was detected was considered to be the decomposition onset temperature.

Differential scanning calorimetry (DSC) analysis was performed using a DSC-3 Mettler-Toledo differential scanning calorimeter (Greifensee, Switzerland) at a heating rate of 10 °C/min in argon.

The efficiency of the network formation was investigated with a gel fraction analysis. Samples were dried to a constant weight in the vacuum oven at 1 mBar and 80 °C. After that, uncross-linked oligomers and precursors were removed by extraction with tetrahydrofuran in a Soxhlet apparatus for 10 h. Finally, the samples were dried in the vacuum oven at 1 mBar and 80 °C until they reached a constant weight. The gel fractions of the samples were calculated by using the following equation:Gel fraction (%) = W_1_/W_0_ × 100
where W_0_ and W_1_ are the dried-gel weights before and after the extraction, respectively.

The mechanical properties of samples were evaluated on a LLOYD Instruments LR5K Plus (Bognor Regis, UK) testing machine with a 100 mm/min stretching speed.

The self-healing properties were studied using a Hitachi TM4000PLUS field-emission scanning electron microscope (Tokyo, Japan) by observing the incised polymer film before and after heating at 150 °C for 2 h.

## 3. Results and Discussion

### 3.1. Preparation

A series of new luminescent cross-linked polymers based on *β*-diketone-modified polysiloxanes and oligoorganoeuropiumsiloxanes were obtained, as illustrated in Figure 1. Polydimethylsiloxane (PDMS) with statistically distributed silylhydride groups **4** was synthesized according to the method described earlier [67,68] through the cationic ring-opening polymerization of a mixture of octamethylcyclotetrasiloxane **1** and methylhydrosiloxane cyclic oligomers **2** using hexamethyldisiloxane **3** as a terminating agent and Amberlyst 15 as a cationic catalyst. Polysiloxane **4** was obtained as a colorless oil with the following molecular characteristics: M_n_ = 29,500 Da, M_w_ = 49,600 Da, PDI = 1.68. The GPC curve of polymer **4** obtained in toluene is presented in Figure 1. The ^1^H NMR spectrum of **4** contained signals that were assigned to the protons of Si-CH_3_ groups (singlet at 0.06 ppm) and the protons of silylhydride groups (multiplet at 4.66–4.68 ppm), and it is presented in Figure 2. The integration of signals in this spectrum gave a ratio of distributed chains (-O-SiHCH_3_- to -O-Si(CH_3_)_2_-) of 1:38. The FTIR spectrum of **4** presented in Figure 3 shows characteristic absorption peaks at 1023–1096 and 801 cm^−1^, which were assigned to asymmetric and symmetric Si-O-Si stretching vibrations, respectively. The peaks at 1264 and 868 cm^−1^ were attributed to the Si-C vibrations, and the peaks at 2911–2969 cm^−1^ were attributed to the C-H vibrations. The low-intensity absorption band at 2160 cm^−1^ was assigned to the Si-H stretching vibration.

The functional ligand Allyl-DBM **5** was prepared with the method described earlier [62]. Using the hydrosilylation reaction between polysiloxane **4** with distributed silylhydride groups and allyl-DBM **5** in toluene in the presence of Karstedt’s catalyst, a polysiloxane **6** with a distributed dibenzoylmethane ligand was obtained as a brownish oil with a good yield. Product **6** was purified through precipitation from toluene by acetonitrile. A GPC analysis of polymer **6** was performed in a toluene solution and gave the following molecular characteristics: M_n_ = 47,000 Da, M_w_ = 123,000 Da, PDI = 2.61 (Figure 4). The increase in the polydispersity index value for polymer **6** was apparently related to the specific interaction of *β*-diketone fragments with the GPC column material, since a similar effect was observed in the case of low-molecular unsubstituted *β*-diketone. The completeness of the reaction was confirmed by the disappearance of the signals assigned to Si-H in the ^1^H NMR (Figure 5) and FTIR spectra (Figure 3) and the appearance of new signals assigned to the propyl-dibenzoylmethane group. In the FTIR spectrum, a new band appeared at 1609 cm^−1^, which was related to C=O stretching vibrations. The ^1^H NMR spectrum contained new signals that corresponded to the protons of propyl fragments (tree triplets at 0.57, 1.70, and 2.69 ppm), the aromatic protons of a dibenzoylmethane fragment (7.27–7.99 ppm), the CH proton of the dicarbonyl group COCHCOH (singlet at 6.84), and the OH proton of the dicarbonyl group COCHCOH in enol form (broadened singlet at 16.95) (Figure 4). The ^29^Si NMR spectrum contained only one signal at −21.92 ppm that corresponded to the dialkyl-siloxy group O-Si(Alk)_2_-O (Figure 4). The absence of other signals in the spectrum, for example, from the silsesquioxane fragment, tells us that the hydrosilylation reaction was carried out without any side reactions. 

Oligoorganoeuropiumsiloxanes **7 a**,**b** were utilized as a source of Eu^3+^ ions for cross-linking through coordination in this work. Compounds **7 a** and **7 b**, which contained phenyl and ethyl groups at the silicon atom, were prepared from phenyltriethoxysilane and ethyltrimethoxysilane, respectively, as well as EuCl_3_ × 6H_2_O or Eu(NO_3_)_3_ × 6H_2_O. Elemental analysis confirmed that there were three silicon atoms for each europium atom and 1.5 oxygen atoms on each silicon atom in **7 a**,**b**. Thus, a common formula of these compounds can be expressed as {[(RSiO_1_._5_)_3_Eu]_n_} × L_x_ (L = n-BuOH and/or H_2_O).

A series of luminescent polymers cross-linked by the interaction between the dibenzoylmethane (DBM) ligands in polymer **6** and Eu^3+^ ions in oligoorganoeuropiumsiloxane were synthesized according to Figure 1. Films from these polymers were obtained after rapidly mixing the precursor solutions in THF, pouring them out on a Teflon plate, and evaporating the solvent. Thus, four luminescent polymer films were obtained from different oligoorganoeuropiumsiloxanes, and they had varying metal-ligand ratios (Table 1, Figure 6).

The FTIR spectra of cross-linked polymers **8 a**–**d** were very similar and are presented in Figure 5. As can be seen in Figure 5, the vibration bands of the C-H bonds were observed at 2905–2969 cm^−1^. The characteristic absorption peaks at 1006–1097 and 790 cm^−1^ were assigned to asymmetric and symmetric Si-O-Si stretching vibrations, respectively, and the peaks at 1256–1262 and 863–868 cm^−1^ were attributed to the Si-C vibrations. The peaks in the region of 1596–1610 cm^−1^ were assigned to C=O vibrations in the coordinated *β*-diketone group.

### 3.2. Characterization

The films obtained from the oligophenyleuropiumsiloxanes (**8 a**,**b**) were opaque, and the films obtained from the oligoethyleuropiumsiloxanes (**8 c**,**d**) were transparent (Figure 6). This may be explained by the increase in the compatibility of ethyl groups with methyl groups in the modified PDMS. In addition, the films containing stoichiometric amounts of Eu^3+^ ions (**8 a**,**c**) were yellow, while the compounds with an excess of metal ions were grayish or almost colorless. This effect could be due to the different degree of complexation of Eu^3+^ ions with the DBM fragment in **6**. The physicochemical properties of all of the synthesized compounds are presented in Table 1.

#### 3.2.1. Gel Fraction

A gel fraction analysis of polymers **8 a**–**d** carried out in tetrahydrofuran showed that all polymers formed stable 3D networks. Polymers **8 a** and **8 b** obtained from oligophenyleuropiumsiloxane **7 a** demonstrated 84 and 79% gel fractions, respectively. Polymer **8 c** obtained from oligoethyleuropiumsiloxane **7 b** showed a decrease in the gel fraction to 53%. The most stable network formed in polymer **8 d**, which had the highest gel fraction of 89%, indicating that it had the most complete cross-linking reaction.

#### 3.2.2. Morphology

A morphological investigation of cross-linked polymer films **8 a**–**d** was carried out using the scanning electron microscopy (SEM) method. Figure 7 shows the SEM micrographs of all prepared films at 1.00k magnification. All samples except **8 b** were almost homogeneous and contained small inclusions that were about 2.5 nm in size in the case of **8 a** and with a size of less than 1 nm in the case of **8 c**,**d**, which could be a result of the crystallization of the excess oligophenyleuropiumsiloxane. In addition, the samples contained small, widely spaced pores with a diameter of about 3 nm, the existence of which can be explained by the incorrect evaporation regime. Apparently, this was related to the condensation of water on the film surface during the fast evaporation of the solvent [69].

### 3.3. Mechanical Properties

The tensile properties of the prepared cross-linked polymer films were studied and are shown in Figure 8 (as stress-strain curves) and in Table 2. As can be seen from the curves, the mechanical properties of the samples depended on both the amount of metal ions and the type of organic radical in oligoorganoeuropiumsiloxane. It is obvious that increasing the amount of Eu^3+^ ions led to an increase in the tensile strength and Young’s modulus. In addition, it should be noted that the films based on oligophenyleuropiumsiloxane (**8 a**,**b**) were more elastic, as they had a higher elongation at break, even with a higher tensile strength.

Polymer **8 a** prepared from oligophenyleuropiumsiloxane (**7 a**), which contained a stoichiometric amount of Eu^3+^ ions, showed a tensile strength of 0.81 MPa and the largest elongation at break ε_p_ (520%) compared to all samples. Polymer **8 b**, which contained twice as much than the stoichiometric amount of Eu^3+^ ions, had a higher Young’s modulus (1.14 MPa) and tensile strength (1.75 MPa), but had a decreased elongation at break (300%), indicating better cross-linking. Polymer **8 c** prepared from oligoethyleuropiumsiloxane (**7 b**), which contained a stoichiometric amount of Eu^3+^ ions, demonstrated a high elongation at break (380%), moderate tensile strength (0.42 MPa), and a Young’s modulus of 0.24 MPa. This decline in mechanical characteristics in **8 c** may have been due to the low stability of the cross-linked 3D network, as determined by the gel fraction analysis. Increasing the amount of oligoethyleuropiumsiloxane in polymer **8 d** to twice as much as the stoichiometric amount led to a four-fold increase in tensile strength (1.74 MPa) and significant growth of the Young’s modulus (1.81 MPa). However, this sample showed the smallest elongation at break (160%). It should be mentioned that this sample exhibited the largest gel fraction, and it is obvious that its considerable cross-linking density improved its mechanical properties.

The full polysiloxane analog that we previously synthesized [67], which we used as a model compound that did not contain metal–*β*-diketonate fragments, had a lower tensile strength (0.15 MPa) and elongation at break (47%) compared to those of polymers **8 a**–**d**. The same can be noted about PDMS cross-linked through nickel *β*-diketonate fragments [67].

Thus, the inclusion of Eu^3+^ ions in **6** led to obtaining luminescent elastic cross-linked polymer films whose mechanical properties could be tuned by varying the amounts of components and their nature.

### 3.4. TGA Analysis

The thermal stability of all samples was estimated in a thermal gravimetric analysis (TGA) in both air and argon in the range from 50 to 700 °C, and data are summarized in Figure 9 and Table 3.

According to the TGA data, the thermal and thermo-oxidative stability of silylhydride-containing polysiloxane **4** was close to the reference data observed for PDMS [70]. The thermo-oxidative stability of polysiloxane **4** slightly decreased after modification with *β*-diketonate fragments (compound **6**); the temperature for 5% weight loss of the samples dropped from 396 to 378 °C, and there was a decrease in the amount of solid residue after the analysis. This seemed to be due to the increase in the amount of the organic component in the polymer.

Oligoorganoeuropiumsiloxanes **7 a**,**b** were thermally unstable and started to lose their weight at the initial temperature of the analysis. The weights of both samples decreased stepwise, indicating the evaporation of the solvent molecules located in the coordination sphere of Eu^3+^ ions and the condensation processes that occurred. The high percentage of the residue after decompression was in agreement with the silsesquioxane structure of the samples.

Cross-linked polymers **8 a**–**d** had a lower temperature at which they started to be destroyed in air and in argon, which could be explained if we suppose that the Eu^3+^ ions that served as cross-linking agents could initiate the rupture of chemical bonds at high temperatures.

### 3.5. DSC Analysis

For all samples, a DSC analysis was carried out. The DSC traces for polymers **4**, **6**, and **8 a**–**d** are presented in Figure 10, and the values of the observed transitions are given in Table 3. As can be seen in Figure 10, the DSC curve for silylhydride-containing polymer **4** contains all transitions that are characteristic for low-molecular-weight polydimethylsiloxane, namely, a heat capacity jump at the glass transition temperature, an exothermic peak of cold crystallization, and a bimodal peak of the melting of the crystalline phase. The introduction of the *β*-diketone fragment into polysiloxane increased the glass transition temperature from −126 to −117 °C. All cross-linked polymer films **8 a**–**d** exhibited glass transition temperatures close to −121 °C, so there was no correlation between the amount of Eu^3+^ ions in the sample and the type of oligoorganoeuropiumsiloxane used for its preparation. It should be noted that, in the DSC curves of compounds **6** and **8 a**–**d,** crystallization and melting transitions were not observed. Thus, it can be concluded that the introduction of *β*-diketone and further cross-linking by Eu^3+^ ions led not only to an increase in the glass transition temperature, but also to a suppression of the crystallization ability of polysiloxane.

### 3.6. Self-Healing Properties

The self-healing properties of the cross-linked polymers **8 a**–**d** were investigated with the SEM method and were attributed to the dynamic nature of the coordination bonds in the Eu^3+^ complexes. Figure 11 demonstrates the self-healing properties of the prepared films after being incised with a scalpel. A thermal treatment at 150 °C for 2 h resulted in an almost complete self-healing of the scratch surface in all studied polymers. It should be mentioned that this process also took place at lower temperatures; however, healing occurred more slowly. Similar thermally triggered self-healing behaviors were observed in other supramolecular metallopolymer networks [6,71,72].

### 3.7. Luminescent Properties

The optical properties of oligoorganoeuropiumsiloxanes **7 a**,**b** and cross-linked polymers **8 a**–**d** were studied in the solid state at room temperature. Figure 12 the emission spectra of compounds **7 a**,**b** and **8 a**–**d** shows normalized at the ^5^D_0_→^7^F_1_ transition. Upon excitation with 300 nm UV light, the emission spectra of all samples showed the five emission peaks that are characteristic for the Eu^3+^ ions arising from the ^5^D_0_ →^7^F_J_ (J = 0–4) transitions. The emission peak at about 590 nm (^5^D_0_ →^7^F_1_) is a magnetic dipole transition, and its intensity is insensitive to the coordination environment, so this transition is used to calibrate the intensity of Eu^3+^ luminescence spectra [73]. Thus, it is possible to compare the luminescence intensities of transitions resulting from the electric dipole ^5^D_0_ →^7^F_2_ transition at about 615 nm, which is sensitive to the surrounding Eu^3+^. A significant luminescence enhancement from this transition comparing to initial oligoorganoeuropiumsiloxanes **7 a**,**b** was observed in coordination polymers **8 a**–**d** due to the coordination of Eu^3+^ with the DBM unit. In addition, there was a correlation between the luminescence intensity of the peak at 615 nm and the nature of the substituent in oligoorganoeuropiumsiloxane: Polymers **8 c**,**d** based on oligoethyleuropiumsiloxane **7 b** demonstrated a higher luminescence intensity of this peak than polymers **8 a**,**c**, based on oligophenyleuropiumsiloxane **7 a**. It should be noted that the luminescence intensity of the peak at 615 nm increased with the better coordination of Eu^3+^ ions: samples **8 a** and **8 c**, which contained more DBM ligands, showed greater intensity than **8 b** and **8 d**, respectively.

The luminescence decay kinetics obtained at 615 nm are presented in Figure 13; they had a non-exponential character. This behavior indicated the presence of Eu^3+^ ions with a variety of coordination shells. As can be seen in Figure 13, the average lifetime increased with the increase in the concentration of Eu^3+^ ions in **8 b**,**d**. Three exponential terms should be used to satisfactorily describe the kinetics of **8 a** and **8 c**. If one supposes that each term is attributed to one type of complex, then the amplitudes of the exponential terms can be used to monitor their concentrations. The results of the joint global fitting of the decays obtained for **8 a**,**b** and **8 c**,**d** are presented in Table 4. In both cases, the increase in oligoorganoeuropiumsiloxane concentration led to a decrease in the fast component with an increase in slow decay.

Figure 14 shows the normalized absorption spectra of polymer **6** in solution and in the solid state, as well as the luminescence excitation spectra of polymers **8 a**–**d** in the solid state with emissions at 615 nm. Comparing the luminescence excitation spectra of the initial metallasiloxanes **7 a**,**b** presented in Figure 15 with those for polymers **8 a**–**d** (Figure 14) provided the reason for explaining that the excitation of the Eu^3+^ ions was mainly through the energy transfer from the DBM ligand.

The maxima of the fluorescence excitation spectra at 290 nm in the case of **8 a** and **8 c** seemed to be related to the inner filter effect due to the high absorption of uncomplexed DBM moieties of **6** in the 300–400 nm range. The yellow color of the **8 a** and **8 c** films in Figure 6 was caused by the same effect. There was a small tail in the solid-state absorption spectra of **6** in the range of 400–425 nm. Upon increasing the concentration of **7 a**,**b** in **8 b** and **8 d**, the amount of free ligand decreased, and the yellow color vanished with the decrease in the fraction of light that was not transferred to Eu^3+^ ions.

It is known that similar complexes possess similar sensory properties to ammonia [74]. Thus, the sensitivity to ammonia of polymer **8 a** was investigated. As can be seen in Figure 16, the presence of ammonia vapors led to a 12-fold increase in the luminescence intensity of polymer **8 a** at an excitation of 370 nm. However, a decrease in emission intensity due to exposure to ammonia was observed if the excitation wavelength was 300 nm, as shown in Figure 16. The luminescence decay kinetics showed a decrease in the average lifetime in the presence of ammonia (Figure 17). The long decay component significantly decreased and almost completely disappeared in the case of **8 a** (see Table 5). As shown by the luminescence decay, only quenching should be observed in the presence of ammonia. Thus, the increase in luminescence observed in the case of excitation at 370 nm was related to the increase in the portion of light absorbed by complexes, as is clearly depicted in Figure 18, where the luminescence excitation spectrum in the range of 300–400 nm sufficiently increased. The observed chemosensory properties are reversible.

## 4. Conclusions

In summary, a series of novel cross-linked luminescent polymers based on multiligand polysiloxanes with grafted *β*-diketone fragments and oligoorganoeuropiumsiloxanes containing different organic substituents were prepared and fully characterized. A gel fraction analysis showed that the prepared coordination polymers formed stable 3D networks with different cross-linking densities. The morphological investigation demonstrated that cross-linked polymer films **8 a**–**d** were almost homogeneous, containing small amounts of inclusions and small pores resulting from the condensation of water on the film surface during fast evaporation of the solvent. Analysis of the stress–strain curves of the cross-linked films showed the dependence of the mechanical properties on both the amount of metal ions and the type of organic substituent in oligoorganoeuropiumsiloxane. An increase in the amount of Eu^3+^ ions led to an increase in tensile strength and Young’s modulus. Polymer films based on oligophenyleuropiumsiloxane (**8 a**,**b**) were more elastic, and films based on oligoethyleuropiumsiloxane (**8 c**,**d**) were more rigid. The Young’s modulus and tensile strength of the prepared cross-linked polymers were much larger than those of the full polysiloxane analog. The thermal properties of all samples were estimated by using thermal gravimetric analysis and the differential scanning calorimetry method. All samples demonstrated a high thermal and thermo-oxidative stability; the temperature at the start of the destruction was found to be about 300 °C. It should be noted that the cross-linked polymers, as well as the polysiloxane modified with *β*-diketone fragments, do not possess crystallization and melting transitions that are characteristic for polysiloxanes, which may be due to difficulties in the packaging of polymer chains with bulky side substituents or cross-links. It was found that the prepared coordination polymers possessed a thermally triggered self-healing behavior. The investigation of the luminescent properties of compounds **7 a**,**b** and **8 a**–**d** showed that there was a significant enhancement of the luminescence intensity of the coordination polymers **8 a**–**d** compared to that of the initial oligoorganoeuropiumsiloxanes **7 a**,**b**, indicating an efficient energy transfer process from the excited state of the ligand to the Eu^3+^ ion. Additionally, these coordination polymers were sensitive to ammonia, as shown by the 12-fold increase in the luminescence intensity upon exposure to ammonia. Thus, a new method for the preparation of elastic luminescent silicone materials with tunable optical and mechanical properties was investigated. Such materials are promising candidates for the design and development of new smart materials that are applicable as luminescent coatings, electronic devices, and sensors.

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
