# Peer review of "Cross-Linked Luminescent Polymers Based on β-Diketone-Modified Polysiloxanes and Organoeuropiumsiloxanes"

_polymers, 2022, doi:10.3390/polym14132554_

Round 1

Reviewer 1 Report

The manuscript presents synthesis and extensive characterization of novel crosslinked luminescent polysiloxane-based materials with Eu3+ ions coordinated to grafted β-diketone groups. The study is interesting, well planned and executed. I recommend publication in the present form.

The only thing I would like to ask for is the expression "polysiloxane with distributed silylhydride (or dibenzoylmethane) groups". It is not precise. Those groups can be regularly distributed or statisticaly/randomly distributed. Please specify the case and additionally provide 29Si NMR spectrum of polymer 4 with appropriate segmental analysis. It is necessary to prove the polymer structure. This information cannot be deduced form the specturm of polymer 6, after the hydrosilylation.

End groups of polymer 6 are not shown in the 29Si spectrum presented in Figure 4. 

Author Response

1. The only thing I would like to ask for is the expression "polysiloxane with distributed silylhydride (or dibenzoylmethane) groups". It is not precise. Those groups can be regularly distributed or statisticaly/randomly distributed.

Answer: SiH groups are statisticaly distributed. It is well known fact that preparation of poly(methylhydrosiloxane) by the cationic ring opening polymerization method lead to obtain statisticaly distributed  poly(methylhydrosiloxane).

It was mentioned in the text.

2. End groups of polymer 6 are not shown in the 29Si spectrum presented in Figure 4.

Answer: End groups are not shown in 29Si spectrum most likely due to large molecular mass of polymer (Mn=47 000 Da, Mw=123 000 Da).

Reviewer 2 Report

This is an interesting manuscript entitled: "Cross-linked luminescent polymers based on β-diketone-modified polysiloxanes and organoeuropiumsiloxanes".

Some editorial corrections are recommended:

1. on pages 2 - a word "organosilicons" should be changed for "organosilicones";

2. on page 4  and 11 - a name "Silylhydride cycles" I propose to change for "methylhydrosiloxane cyclic oligomers";

3. on page 4 - abbreviations "DBM" and "Allyl-DBM" should be explained;

4. on page 5 - in reaction of EtSi(OMe)3 with NaOH not only cis-tetraethylcyclotetrasiloxanolate was formed, but also other higher cyclic oligosiloxanes (n 4) in lower yields;

5. terms "the largest deformation at the breaking εÑ€" (on page 14) and "Strain at break" (in Table 2) should be changed for "elongation at break εÑ€";

6. the term "the temperature of start the destruction" (on p. 17) should be changed for "the temperature of 5% weight loss of samples";

7. the term "moisture condensation during solvent evaporation" (on p. 25) sounds strange;

8. the term "low-weight polydimethylsiloxane" (on p. 17) should be changed for "low-molecular weight polydimethylsiloxane".

9. Other "small" English errors should be corrected.

Author Response

1. on pages 2 - a word "organosilicons" should be changed for "organosilicones";

Answer: it was corrected.

2. on page 4  and 11 - a name "Silylhydride cycles" I propose to change for "methylhydrosiloxane cyclic oligomers";

Answer: it was corrected.

3. on page 4 - abbreviations "DBM" and "Allyl-DBM" should be explained;

Answer: done.

4. on page 5 - in reaction of EtSi(OMe)3 with NaOH not only cis-tetraethylcyclotetrasiloxanolate was formed, but also other higher cyclic oligosiloxanes (n ≥ 4) in lower yields;

Answer: it was corrected.

5. terms "the largest deformation at the breaking εÑ€" (on page 14) and "Strain at break" (in Table 2) should be changed for "elongation at break εÑ€";

Answer: it was corrected.

6. the term "the temperature of start the destruction" (on p. 17) should be changed for "the temperature of 5% weight loss of samples";

Answer: it was corrected.

7. the term "moisture condensation during solvent evaporation" (on p. 25) sounds strange;

Answer: it was corrected on "condensation of water on the film surface during fast solvent evaporation".

8. the term "low-weight polydimethylsiloxane" (on p. 17) should be changed for "low-molecular weight polydimethylsiloxane".

Answer: it was corrected.

Reviewer 3 Report

A new method for the synthesis of elastic luminescent silicone materials with tuned optical and mechanical properties has been developed. The synthesized polymer objects are promising as luminescent coatings in electronic devices and sensors. New luminescent materials have been studied in detail by modern physicochemical methods. The results are discussed in detail at a high scientific level. I think that the article is worthy of publication in Polymers.

Author Response

Thank you very much!